# GENERALIZATION AND ESTIMATION ERROR BOUNDS FOR MODEL-BASED NEURAL NETWORKS

**Avner Shultzman[1], Eyar Azar[1], Miguel R. D. Rodrigues[2] & Yonina C. Eldar[1]** [*]
Faculty of Mathematics and Computer Science, Weizmann Institute of Science, Israel[1]
Faculty of Engineering Science, University College London, UK[2]
`{avner.shultzman,eyar.azar,yonina.eldar}@weizmann.ac.il`
`m.rodrigues@ucl.ac.uk`

## ABSTRACT

Model-based neural networks provide unparalleled performance for various tasks, such as sparse coding and compressed sensing problems. Due to the strong connection with the sensing model, these networks are interpretable and inherit prior structure of the problem. In practice, model-based neural networks exhibit higher generalization capability compared to ReLU neural networks. However, this phenomenon was not addressed theoretically. Here, we leverage complexity measures including the global and local Rademacher complexities, in order to provide upper bounds on the generalization and estimation errors of model-based networks. We show that the generalization abilities of model-based networks for sparse recovery outperform those of regular ReLU networks, and derive practical design rules that allow to construct model-based networks with guaranteed high generalization. We demonstrate through a series of experiments that our theoretical insights shed light on a few behaviours experienced in practice, including the fact that ISTA and ADMM networks exhibit higher generalization abilities (especially for small number of training samples), compared to ReLU networks.

## 1 INTRODUCTION

Model-based neural networks provide unprecedented performance gains for solving sparse coding problems, such as the learned iterative shrinkage and thresholding algorithm (ISTA) (Gregor & LeCun, 2010) and learned alternating direction method of multipliers (ADMM) (Boyd et al., 2011). In practice, these approaches outperform feed-forward neural networks with ReLU nonlinearities.

These neural networks are usually obtained from algorithm unrolling (or unfolding) techniques, which were first proposed by Gregor and LeCun (Gregor & LeCun, 2010), to connect iterative algorithms to neural network architectures. The trained networks can potentially shed light on the problem being solved. For ISTA networks, each layer represents an iteration of a gradient-descent procedure. As a result, the output of each layer is a valid reconstruction of the target vector, and we expect the reconstructions to improve with the network's depth. These networks capture original problem structure, which translates in practice to a lower number of required training data (Monga et al., 2021). Moreover, the generalization abilities of model-based networks tend to improve over regular feed-forward neural networks (Behboodi et al., 2020; Schnoor et al., 2021).

Understanding the generalization of deep learning algorithms has become an important open question. The generalization error of machine learning models measures the ability of a class of estimators to generalize from training to unseen samples, and avoid overfitting the training (Jakubovitz et al., 2019). Surprisingly, various deep neural networks exhibit high generalization abilities, even for increasing networks' complexities (Neyshabur et al., 2015b; Belkin et al., 2019). Classical machine learning

---

[*]This project has received funding from the European Research Council (ERC) under the European Union's Horizon 2020 research, the innovation programme (grant agreement No. 101000967), and the Israel Science Foundation under Grant 536/22. Y. C. Eldar and M. R. D. Rodrigues are supported by The Weizmann-UK Making Connections Programme (Ref. 129589). M. R. D. Rodrigues is also supported by the Alan Turing Institute. The authors wish to thank Dr. Gholamali Aminian from the Alan Turing Institute, UK, for his contribution to the proofs' correctness.

measures such as the Vapnik-Chervonenkis (VC) dimension (Vapnik & Chervonenkis, 1991) and Rademacher complexity (RC) (Bartlett & Mendelson, 2002), predict an increasing generalization error (GE) with the increase of the models' complexity, and fail to explain the improved generalization observed in experiments. More advanced measures consider the training process and result in tighter bounds on the estimation error (EE), were proposed to investigate this gap, such as the local Rademacher complexity (LRC) (Bartlett et al., 2005). To date, the EE of model based networks using these complexity measures has not been investigated to the best of our knowledge.

## 1.1 OUR CONTRIBUTIONS

In this work, we leverage existing complexity measures such as the RC and LRC, in order to bound the generalization and estimation errors of learned ISTA and learned ADMM networks.

- We provide new bounds on the GE of ISTA and ADMM networks, showing that the GE of model-based networks is lower than that of the common ReLU networks. The derivation of the theoretical guarantees combines existing proof techniques for computing the generalization error of multilayer networks with new methodology for bounding the RC of the soft-thresholding operator, that allows a better understanding of the generalization ability of model based networks.

- The obtained bounds translate to practical design rules for model-based networks which guarantee high generalization. In particular, we show that a nonincreasing GE as a function of the network's depth is achievable, by limiting the weights' norm in the network. This improves over existing bounds, which exhibit a logarithmic increase of the GE with depth (Schnoor et al., 2021). The GE bounds of the model-based networks suggest that under similar restrictions, learned ISTA networks generalize better than learned ADMM networks.

- We also exploit the LRC machinery to derive bounds on the EE of feed-forward networks, such as ReLU, ISTA, and ADMM networks. The EE bounds depend on the data distribution and training loss. We show that the model-based networks achieve lower EE bounds compared to ReLU networks.

- We focus on the differences between ISTA and ReLU networks, in term of performance and generalization. This is done through a series of experiments for sparse vector recovery problems. The experiments indicate that the generalization abilities of ISTA networks are controlled by the soft-threshold value. For a proper choice of parameters, ISTA achieves lower EE along with more accurate recovery. The dependency of the EE as a function of $\lambda$ and the number of training samples can be explained by the derived EE bounds.

## 1.2 RELATED WORK

Understanding the GE and EE of general deep learning algorithms is an active area of research. A few approaches were proposed, which include considering networks of weights matrices with bounded norms (including spectral and $L_{2,1}$ norms) (Bartlett et al., 2017; Sokolić et al., 2017), and analyzing the effect of multiple regularizations employed in deep learning, such as weight decay, early stopping, or drop-outs, on the generalization abilities (Neyshabur et al., 2015a; Gao & Zhou, 2016; Amjad et al., 2021). Additional works consider global properties of the networks, such as a bound on the product of all Frobenius norms of the weight matrices in the network (Golowich et al., 2018). However, these available bounds do not capture the GE behaviour as a function of network depth, where an increase in depth typically results in improved generalization. This also applies to the bounds on the GE of ReLU networks, detailed in Section 2.3.

Recently, a few works focused on bounding the GE specifically for deep iterative recovery algorithms (Behboodi et al., 2020; Schnoor et al., 2021). They focus on a broad class of unfolded networks for sparse recovery, and provide bounds which scale logarithmically with the number of layers (Schnoor et al., 2021). However, these bounds still do not capture the behaviours experienced in practice.

Much work has also focused on incorporating the networks' training process into the bounds. The LRC framework due to Bartlett, Bousquet, and Mendelson (Bartlett et al., 2005) assumes that the training process results in a smaller class of estimation functions, such that the distance between the estimator in the class and the empirical risk minimizer (ERM) is bounded. An additional related framework is the effective dimensionality due to Zhang (Zhang, 2002). These frameworks result in

different bounds, which relate the EE to the distance between the estimators. These local complexity measures were not applied to model-based neural networks.

Throughout the paper we use boldface lowercase and uppercase letters to denote vectors and matrices respectively. The $L_1$ and $L_2$ norms of a vector $\boldsymbol{x}$ are written as $\|\boldsymbol{x}\|_1$ and $\|\boldsymbol{x}\|_2$ respectively, and the $L_\infty$ (which corresponds to the maximal $L_1$ norm over the matrix's rows) and spectral norms of a matrix $\boldsymbol{X}$, are denoted by $\|\boldsymbol{X}\|_\infty$ and $\|\boldsymbol{X}\|_\sigma$ respectively. We denote the transpose operation by $(\cdot)^T$. For any function $f$ and class of functions $\mathcal{H}$, we define $f \circ \mathcal{H} = \{x \mapsto f \circ h(x) : h \in \mathcal{H}\}$.

## 2 PRELIMINARIES

### 2.1 NETWORK ARCHITECTURE

We focus on model-based networks for sparse vector recovery, applicable to the linear inverse problem

$$\boldsymbol{y} = \boldsymbol{A}\boldsymbol{x} + \boldsymbol{e} \tag{1}$$

where $\boldsymbol{y} \in \mathbb{R}^{n_y}$ is the observation vector with $n_y$ entries, $\boldsymbol{x} \in \mathbb{R}^{n_x}$ is the target vector with $n_x$ entries, with $n_x > n_y$, $\boldsymbol{A} \in \mathbb{R}^{n_y \times n_x}$ is the linear operator, and $\boldsymbol{e} \in \mathbb{R}^{n_y}$ is additive noise. The target vectors are sparse with sparsity rate $\rho$, such that at most $\lfloor \rho n_x \rfloor$ entries are nonzero. The inverse problem consists of recovering the target vector $\boldsymbol{x}$, from the observation vector $\boldsymbol{y}$.

Given that the target vector is assumed to be sparse, recovering $\boldsymbol{x}$ from $\boldsymbol{y}$ in (1) can be formulated as an optimization problem, such as least absolute shrinkage and selection operator (LASSO) (Tibshirani & Ryan, 2013), that can be solved with well-known iterative methods including ISTA and ADMM. To address more complex problems, such as an unknown linear mapping $\boldsymbol{A}$, and to avoid having to fine tune parameters, these algorithms can be mapped into model-based neural networks using unfolding or unrolling techniques (Gregor & LeCun, 2010; Boyd et al., 2011; Monga et al., 2021; Yang et al., 2018). The network's architecture imitates the original iterative method's functionality and enables to learn the models' parameters with respect to a set of training examples.

We consider neural networks with $L$ layers (referred to as the network's depth), which corresponds to the number of iterations in the original iterative algorithm. The layer outputs of an unfolded ISTA network $\boldsymbol{h}_I^l$, $l \in [1, L]$, are defined by the following recurrence relation, shown in Fig. 1:

$$\boldsymbol{h}_I^l = \mathcal{S}_\lambda \left( \boldsymbol{W}^l \boldsymbol{h}_I^{l-1} + \boldsymbol{b} \right), \quad \boldsymbol{h}_I^0 = \mathcal{S}_\lambda(\boldsymbol{b}) \tag{2}$$

where $\boldsymbol{W}^l \in \mathbb{R}^{n_x \times n_x}$, $l \in [1, L]$ are the weights matrices corresponding to each of the layers, with bounded norms $\|\boldsymbol{W}^l\|_\infty \leq B_l$. We further assume that the $L_2$ norm of $\boldsymbol{W}^1$ is bounded by $B_1$. The vector $\boldsymbol{b} = \boldsymbol{A}^T \boldsymbol{y}$ is a constant bias term that depends on the observation $\boldsymbol{y}$, where we assume that the initial values are bounded, such that $\|\boldsymbol{h}_I^0\|_1 \leq B_0$. In addition, $\mathcal{S}_\lambda(\cdot)$ is the elementwise soft-thresholding operator

$$\mathcal{S}_\lambda(\boldsymbol{h}) = \text{sign}(\boldsymbol{h}) \max(|\boldsymbol{h}| - \lambda, \boldsymbol{0}) \tag{3}$$

where the functions $\text{sign}(\cdot)$ and $\max(\cdot)$ are applied elementwise, and $\boldsymbol{0}$ is a vector of zeros. As $\mathcal{S}_\lambda(\cdot)$ is an elementwise function it preserves the input's dimension, and can be applied on scalar or vector inputs. The network's prediction is given by the last layer in the network $\hat{\boldsymbol{x}} = \boldsymbol{h}_I^L(\boldsymbol{y})$. We note that the estimators are functions mapping $\boldsymbol{y}$ to $\hat{\boldsymbol{x}}$, $\boldsymbol{h}_I^L : \mathbb{R}^{n_y} \to \mathbb{R}^{n_x}$, characterized by the weights, i.e. $\boldsymbol{h}_I^L = \boldsymbol{h}_I^L(\{\boldsymbol{W}^l\}_{l=1}^L)$. The class of functions representing the output at depth $L$ in an ISTA network, is $\mathcal{H}_I^L = \{\boldsymbol{h}_I^L(\{\boldsymbol{W}^l\}_{l=1}^L) : \|\boldsymbol{W}^l\|_\infty \leq B_l \ l \in [1, L], \|\boldsymbol{W}^1\|_2 \leq B_1\}$.

Similarly, the $l$th layer of unfolded ADMM is defined by the following recurrence relation

$$\begin{aligned} \boldsymbol{h}_A^l &= \boldsymbol{W}^l \left( \boldsymbol{z}^{l-1} + \boldsymbol{u}^{l-1} \right) + \boldsymbol{b} \\ \boldsymbol{z}^l &= \mathcal{S}_\lambda \left( \boldsymbol{h}_A^l - \boldsymbol{u}^{l-1} \right), \qquad \boldsymbol{z}^0 = \boldsymbol{0} \\ \boldsymbol{u}^l &= \boldsymbol{u}^{l-1} - \gamma \left( \boldsymbol{h}_A^l - \boldsymbol{z}^l \right), \qquad \boldsymbol{u}^0 = \boldsymbol{0} \end{aligned} \tag{4}$$

where $\boldsymbol{0}$ is a vector of zeros, $\boldsymbol{b} = \boldsymbol{A}^T \boldsymbol{y}$ is a constant bias term, and $\gamma > 0$ is the step size derived by the original ADMM algorithm, as shown in Fig. 1. The estimators satisfy $\boldsymbol{h}_A^L : \mathbb{R}^{n_y} \to \mathbb{R}^{n_x}$, and the class of functions representing the output at depth $L$ in an ADMM network, is $\mathcal{H}_A^L = \{\boldsymbol{h}_A^L(\{\boldsymbol{W}^l\}_{l=1}^L) : \|\boldsymbol{W}^l\|_\infty \leq B_l \ l \in [1, L], \|\boldsymbol{W}^1\|_2 \leq B_1\}$, where we impose the same assumptions on the weights matrices as ISTA networks.

For a depth-$L$ ISTA or ADMM network, the learnable parameters are the weight matrices $\{\boldsymbol{W}^l\}_{l=1}^L$. The weights are learnt by minimizing a loss function $\mathcal{L}$ on a set of $m$ training examples $S = \{(\boldsymbol{x}_i, \boldsymbol{y}_i)\}_{i=1}^m$, drawn from an unknown distribution $\mathcal{D}$, consistent with the model in (1). We consider the case where the per-example loss function is obtained by averaging over the example per-coordinate losses:

$$\mathcal{L}\left(\boldsymbol{h}(\boldsymbol{y}), \boldsymbol{x}\right) = \frac{1}{n_x} \sum_{j=1}^{n_x} \ell\left(h_j(\boldsymbol{y}), \boldsymbol{x}_j\right) \tag{5}$$

where $h_j(\boldsymbol{y})$ and $\boldsymbol{x}_j$ denote the $j$th coordinate of the estimated and true targets, and $\ell : \mathbb{R} \times \mathbb{R} \to \mathbb{R}_+$ is 1-Lipschitz in its first argument. This requirement is satisfied in many practical settings, for example with the $p$-power of $L_p$ norms, and is also required in a related work (Xu et al., 2016). The loss of an estimator $\boldsymbol{h} \in \mathcal{H}$ which measures the difference between the true value $\boldsymbol{x}$ and the estimation $\boldsymbol{h}(\boldsymbol{y})$, is denoted for convenience by $\mathcal{L}(\boldsymbol{h}) = \mathcal{L}\left(\boldsymbol{h}(\boldsymbol{y}), \boldsymbol{x}\right)$.

There exists additional forms of learned ISTA and ADMM networks, which include learning an additional set of weight matrices affecting the bias terms (Monga et al., 2021). Also, the optimal value of $\lambda$ generally depends on the target vector sparsity level. Note however that for learned networks, the value of $\lambda$ at each layer can also be learned. However, here we focus on a more basic architecture with fixed $\lambda$ in order to draw theoretical conclusions.

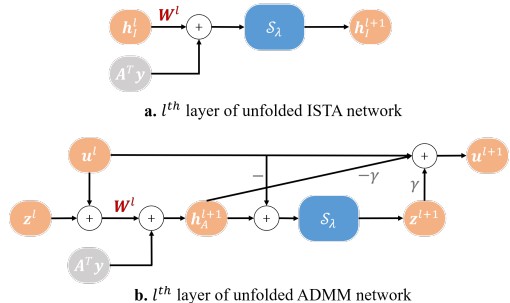

**a.** $l^{th}$ layer of unfolded ISTA network

**b.** $l^{th}$ layer of unfolded ADMM network

Figure 1: A single layer in the unfolded networks. **a.** Unfolded ISTA. **b.** Unfolded ADMM. The learnable parameters are the weight matrices (marked in red).

## 2.2 GENERALIZATION AND ESTIMATION ERRORS

In this work, we focus on upper bounding the GE and EE of the model-based neural networks of Fig. 1. The GE of a class of estimation functions $\boldsymbol{h} \in \mathcal{H}$, such that $\boldsymbol{h} : \mathbb{R}^{n_y} \to \mathbb{R}^{n_x}$, is defined as

$$\mathcal{G}(\mathcal{H}) = \mathbb{E}_S \sup_{\boldsymbol{h} \in \mathcal{H}} L_{\mathcal{D}}(\boldsymbol{h}) - L_S(\boldsymbol{h}) \tag{6}$$

where $L_{\mathcal{D}}(\boldsymbol{h}) = \mathbb{E}_{\mathcal{D}} \mathcal{L}(\boldsymbol{h})$ is the expected loss with respect to the data distribution $\mathcal{D}$ (the joint probability distribution of the targets and observations), $L_S(\boldsymbol{h}) = \frac{1}{m} \sum_{i=1}^m \mathcal{L}(\boldsymbol{h}(\boldsymbol{y}_i), \boldsymbol{x}_i)$ is the average empirical loss with respect to the training set $S$, and $\mathbb{E}_S$ is the expectation over the training datasets. The GE is a global property of the class of estimators, which captures how the class of estimators $\mathcal{H}$ is suited to the learning problem. Large GE implies that there are hypotheses in $\mathcal{H}$ for which $L_D$ deviates much from $L_S$, on average over $S$. However, the GE in (6) does not capture how the learning algorithm chooses the estimator $\boldsymbol{h} \in \mathcal{H}$.

In order to capture the effect of the learning algorithm, we consider local properties of the class of estimators, and focus on bounding the estimation error (EE)

$$\mathcal{E}(\mathcal{H}) = L_{\mathcal{D}}\left(\hat{\boldsymbol{h}}\right) - \inf_{\boldsymbol{h} \in \mathcal{H}} L_{\mathcal{D}}(\boldsymbol{h}) \tag{7}$$

where $\hat{\boldsymbol{h}}$ is the ERM satisfying $L_S(\hat{\boldsymbol{h}}) = \inf_{\boldsymbol{h} \in \mathcal{H}} L_S(\boldsymbol{h})$. We note that the ERM approximates the learned estimator $\boldsymbol{h}_{learned}$, which is obtained by training the network on a set of training examples $S$, using algorithms such as SGD. However, the estimator $\boldsymbol{h}_{learned}$ depends on the optimization algorithm, and can differ from the ERM. The difference between the empirical loss associated with $\hat{\boldsymbol{h}}$ and the empirical loss associated with $\boldsymbol{h}_{learned}$ is usually referred to as the optimization error.

Common deep neural network architectures have large GE compared to the low EE achieved in practice. This still holds, when the networks are not trained with explicit regularization, such as weight decay, early stopping, or drop-outs (Srivastava et al., 2014; Neyshabur et al., 2015b). This empirical phenomena is experienced across various architectures and hyper-parameter choices (Liang et al., 2019; Novak et al., 2018; Lee et al., 2018; Neyshabur et al., 2018).

## 2.3 RADEMACHER COMPLEXITY BASED BOUNDS

The RC is a standard tool which captures the ability of a class of functions to approximate noise, where a lower complexity indicates a reduced generalization error. Formally, the empirical RC of a class of scalar estimators $\mathcal{H}$, such that $h : \mathbb{R}^{n_y} \to \mathbb{R}$ for $h \in \mathcal{H}$, over $m$ samples is

$$\mathfrak{R}_m\left(\mathcal{H}\right) = \mathbb{E}_{\{\epsilon_i\}_{i=1}^m} \sup_{h \in \mathcal{H}} \frac{1}{m} \sum_{i=1}^m \epsilon_i h(\boldsymbol{y}_i) \tag{8}$$

where $\{\epsilon_i\}_{i=1}^m$ are independent Rademacher random variables for which $Pr\left(\epsilon_i = 1\right) = Pr\left(\epsilon_i = -1\right) = 1/2$, the samples $\{\boldsymbol{y}_i\}_{i=1}^m$ are obtained by the model in (1) from $m$ i.i.d. target vectors $\{\boldsymbol{x}_i\}_{i=1}^m$ drawn from an unknown distribution. Taking the expectation of the RC of $\mathcal{L} \circ \mathcal{H}$ with respect to the set of examples $S$ presented in Section 2.1, leads to a bound on the GE

$$\mathcal{G}(\mathcal{H}) \leq 2\mathbb{E}_S \mathfrak{R}_m\left(\mathcal{L} \circ \mathcal{H}\right) \tag{9}$$

where $\mathcal{L}$ is defined in Section 2.1 and $\mathcal{L} \circ \mathcal{H} = \{(\boldsymbol{x}, \boldsymbol{y}) \mapsto \mathcal{L}(h(\boldsymbol{y}), \boldsymbol{x}) : h \in \mathcal{H}\}$ (Shalev-Shwartz & Ben-David, 2014). We observe that the class of functions $\mathcal{L} \circ \mathcal{H}$ consists of scalar functions, such that $f : \mathbb{R}^{n_x} \times \mathbb{R}^{n_x} \to \mathbb{R}$ for $f \in \mathcal{L} \circ \mathcal{H}$. Therefore, in order to bound the GE of the class of functions defined by ISTA and ADMM networks, we can first bound their RC.

Throughout the paper, we compare the model-based architectures with a feed forward network with ReLU activations, given by $ReLU(\boldsymbol{h}) = \max\left(\boldsymbol{h}, \boldsymbol{0}\right)$. In this section, we review existing bounds for the generalization error of these networks. The layers of a ReLU network $\boldsymbol{h}_R^l$, $\forall l \in [1, L]$, are defined by the following recurrence relation $\boldsymbol{h}_R^l = ReLU\left(\boldsymbol{W}^l \boldsymbol{h}_R^{l-1} + \boldsymbol{b}\right)$, and $\boldsymbol{h}_R^0 = ReLU(\boldsymbol{b})$, where $\boldsymbol{W}^l$, $\forall l \in [1, L]$ are the weight matrices which satisfy the same conditions as the weight matrices of ISTA and ADMM networks. The class of functions representing the output at depth $L$ in a ReLU network, is $\mathcal{H}_R^L = \{\boldsymbol{h}_R^L(\{\boldsymbol{W}^l\}_{l=1}^L) : ||\boldsymbol{W}^l||_\infty \leq B_l \ l \in [1, L], \ ||\boldsymbol{W}^1||_2 \leq B_1\}$. This architecture leads to the following bound on the GE.

**Theorem 1** (Generalization error bound for ReLU networks (Gao & Zhou, 2016))**.** *Consider the class of feed forward networks of depth-L with ReLU activations, $\mathcal{H}_R^L$, as described in Section 2.3, and $m$ i.i.d. training samples. Given a $1$-Lipschitz loss function, its GE satisfies $\mathcal{G}\left(\mathcal{H}_R^L\right) \leq 2G_R^l$, where $G_R^l = \frac{B_0 \prod_{l=1}^L B_l}{\sqrt{m}}$.*

*Proof.* Follows from applying the bound of the RC of ReLU neural networks from (Gao & Zhou, 2016) and combining it with (9). □

The bound in Theorem 1 is satisfied for any feed forward network with $1$-Lipschitz nonlinear activations (including ReLU), and can be generalized for networks with activations with different Lipshcitz constants. We show in Theorem 2, that the bound presented in Theorem 1 cannot be substantially improved for ReLU networks with the RC framework.

**Theorem 2** (Lower Rademacher complexity bound for ReLU networks (Bartlett et al., 2017))**.** *Consider the class of feed forward networks of depth-L with ReLU activations, where the weight matrices have bounded spectral norm $||\boldsymbol{W}^l||_\sigma \leq B_l'$, $l \in [1, L]$. The dimension of the output layer is $1$, and the dimension of each non-output layer is at least $2$. Given $m$ i.i.d. training samples, there exists a $c$ such that $\mathfrak{R}_m\left(\mathcal{H}_R'^{,L}\right) \geq cB_0 \prod_{l=1}^L B_l'$, where $\mathcal{H}_R'^{,L} = \{h_R^L(\{\boldsymbol{W}^l\}_{l=1}^L) : ||\boldsymbol{W}^l||_\sigma \leq B_l' \ l \in [1, L]\}$.*

This result shows that using the RC framework the GE of ReLU networks behaves as the product of the weight matrices' norms $\prod_{l=1}^L B_l$, as captured in Theorem 1. Theorem 2, implies that the dependence on the weight matrices' norms, cannot be substantially improved with the RC framework for ReLU networks.

## 3 Generalization Error Bounds: Global Properties

In this section, we derive theoretical bounds on the GE of learned ISTA and ADMM networks. From these bounds we deduce design rules to construct ISTA and ADMM networks with a GE which does not increase exponentially with the number of layers. We start by presenting theoretical guarantees on the RC of any class of functions, after applying the soft-thresholding operation.

Soft-thresholding is a basic block that appears in multiple iterative algorithms, and therefore is used as the nonlinear activation in many model-based networks. It results from the proximal gradient of the $L_1$ norm (Palomar & Eldar, 2010). We therefore start by presenting the following lemma which expresses how the RC of a class of functions is affected by applying soft-thresholding to each function in the class. The proof is provided in the supplementary material.

**Lemma 1** (Rademacher complexity of soft-thresholding). *Given any class of scalar functions $\mathcal{H}$ where $h : \mathbb{R}^n \to \mathbb{R}$, $h \in \mathcal{H}$ for any integer $n \geq 1$, and $m$ i.i.d. training samples,*

$$\mathfrak{R}_m \left( S_\lambda \circ \mathcal{H} \right) \leq \mathfrak{R}_m \left( \mathcal{H} \right) - \frac{\lambda T}{m}, \tag{10}$$

*where* $T = \sum_{i=1}^m T_i$ *and* $T_j = \mathbb{E}_{\{\epsilon_i\}_{i=2}^m} \left( \mathbb{1}_{h^\star(\boldsymbol{y}_j) > \lambda \wedge h'^\star(\boldsymbol{y}_j) < -\lambda} \right)$ *such that* $h^*, h'^* = \arg\max_{h,h' \in \mathcal{H}} \left( h(\boldsymbol{y}_1) - h'(\boldsymbol{y}_1) - 2\lambda \mathbb{1}_{h(\boldsymbol{y}_1) > \lambda \wedge h'(\boldsymbol{y}_1) < -\lambda} + \sum_{i=2}^m \epsilon_i(\mathcal{S}_\lambda(h(\boldsymbol{y}_i)) + \mathcal{S}_\lambda(h'(\boldsymbol{y}_i))) \right).$

The quantity $T$ is a non-negative value obtained during the proof, which depends on the networks' number of layers, underlying data distribution $\mathcal{D}$ and soft-threshold value $\lambda$. As seen from (10), the value of $T$ dictates the reduction in RC due to soft-thresholding, where a reduction in the RC can also be expected. The value of $T$ increases as $\lambda$ decreases. In the case that $\lambda T$ increases with $\lambda$, higher values of $\lambda$ further reduce the RC of the class of functions $\mathcal{H}$, due to the soft-thresholding.

We now focus on the class of functions representing the output of a neuron at depth $L$ in an ISTA network, $\mathcal{H}_I^L$. In the following theorem, we bound its GE using the RC framework and Lemma 1. The proof is provided in the supplementary material.

**Theorem 3** (Generalization error bound for ISTA networks). *Consider the class of learned ISTA networks of depth $L$ as described in (2), and $m$ i.i.d. training samples. Then there exist $T^{(l)}$ for $l \in [1, L]$ in the range $T^{(l)} \in \left[ 0, \min \left\{ \frac{m B_l G_I^{l-1}}{\lambda}, m \right\} \right]$ such that $\mathcal{G}\left( \mathcal{H}_I^L \right) \leq 2 \mathbb{E}_S G_I^L$, where*

$$G_I^l = \left( \frac{B_0 \prod_{l'=1}^l B_{l'}}{\sqrt{m}} - \frac{\lambda}{m} \sum_{l'=1}^{l-1} T^{(l')} \prod_{j=l'+1}^l B_j - \frac{\lambda T^{(l)}}{m} \right). \tag{11}$$

Next, we show that for a specific distribution, the expected value of $T^{(l)}$ is greater than $0$. Under an additional bound on the expectation value of the estimators (specified in the supplementary material)

$$\mathbb{E}_S \left( T^{(l)} \right) \geq \max \left\{ m \left( 1 - 2e^{-(cB_l b^{(l)} - \lambda)} \right), 0 \right\} \tag{12}$$

where $b^{(l+1)} = B_l b^{(l)} - \lambda$, $b^{(1)} = B_0$, and $c \in (0, 1]$. Increasing $\lambda$ or decreasing $B$, will decrease the bound in (12), since crossing the threshold is less probable. Depending on $\lambda$ and $\{B_l\}_{l=1}^L$ (specifically that $\lambda \leq cB_l b^{(l)} + \ln 2$), the bound in (12) is positive, and enforces a non-zero reduction in the GE. Along with Theorem 3, this shows the expected reduction in GE of ISTA networks compared to ReLU netowrks. The reduction is controlled by the value of the soft threshold.

To obtain a more compact relation, we can choose the maximal matrices' norm $B = \max_{l \in [1,L]} B_l$, and denote $T = \min_{l \in [1,L]} T^{(l)} \in [0, m]$ which leads to $\mathcal{G}\left( \mathcal{H}_I^L \right) \leq 2 \left( \frac{B_0 B^L}{\sqrt{m}} - \frac{\lambda \mathbb{E}_S(T)}{m} \frac{B^L - 1}{B - 1} \right)$.

Comparing this bound with the GE bound for ReLU networks presented in Theorem 1, shows the expected reduction due to the soft thresholding activation. This result also implies practical rules for designing low generalization error ISTA networks. We note that the network's parameters such as the soft-threshold value $\lambda$ and number of samples $m$, are predefined by the model being solved (for example, in ISTA, the value of $\lambda$ is chosen according to the singular values of $\boldsymbol{A}$).

We derive an implicit design rule from (3), for a nonincreasing GE, as detailed in Section A.2. This is done by restricting the matrices' norm to satisfy $B \leq 1 + \frac{\lambda \mathbb{E}_S(T)}{\sqrt{m} B_0}$. Moreover, these results can

be extended to convolutional neural networks. As convolution operations can be expressed via multiplication with a convolution matrix, the presented results are also satisfied in that case.

Similarly, we bound the GE of the class of functions representing the output at depth $L$ in an ADMM network, $\mathcal{H}_A^L$. The proof and discussion are provided in the supplementary material.

**Theorem 4** (Generalization error bound for ADMM networks). *Consider the class of learned ADMM networks of depth $L$ as described in (4), and $m$ i.i.d. training samples. Then there exist $T^{(l)}$ for $l \in [1, L-1]$ in the interval $T^{(l)} \in \left[0, \min\left\{\frac{m\tilde{B}_l G_A^{l-1}}{\tilde{\lambda}}, m\right\}\right]$ where $G_A^l = \frac{B_0 \prod_{l'=1}^{l-1} \tilde{B}_{l'}}{\sqrt{m}} - \frac{\tilde{\lambda}}{m} \sum_{l'=1}^{l-2} T^{(l')} \prod_{j=l'+1}^{l-1} \tilde{B}_j - \frac{\tilde{\lambda} T^{(l-1)}}{m}$, and $\tilde{\lambda} = (1+\gamma)\lambda$, $\tilde{B}_l = (1+2\gamma)(B_l + 2)$, $l \in [1, L]$, such that $\mathcal{G}\left(\mathcal{H}_A^L\right) \leq 2\tilde{B}_L \mathbb{E}_S G_A^{L-1}$.*

We compare the GE bounds for ISTA and ADMM networks, to the bound on the GE of ReLU networks presented in Theorem 1. We observe that both model-based networks achieve a reduction in the GE, which depends on the soft-threshold, the underlying data distribution, and the bound on the norm of the weight matrices. Following the bound, we observe that the soft-thresholding nonlinearity is most valuable in the case of small number of training samples. The soft-thresholding nonlinearity is the key that enables reducing the GE of the ISTA and ADMM networks compared to ReLU networks. Next, we focus on bounding the EE of feed-forward networks based on the LRC framework.

## 4 ESTIMATION ERROR BOUNDS: LOCAL PROPERTIES

To investigate the model-based networks' EE, we use the LRC framework (Bartlett et al., 2005). Instead of considering the entire class of functions $\mathcal{H}$, the LRC considers only estimators which are close to the optimal estimator $\mathcal{H}_r = \left\{ h \in \mathcal{H} : \mathbb{E}_\mathcal{D} \|h(y) - h^*(y)\|_2^2 \leq r \right\}$, where $h^*$ is such that $L_D(h^*) = \inf_{h \in \mathcal{H}} L_\mathcal{D}(h)$. It is interesting to note that the class of estimators $\mathcal{H}_r$ only restricts the distance between the estimators themselves, and not between their corresponding losses. Following the LRC framework, we consider target vectors ranging in $[-1, 1]^{n_x}$. Therefore, we adapt the networks' estimations by clipping them to lie in the interval $[-1, 1]^{n_x}$. In our case we consider the restricted classes of functions representing the output of a neuron at depth $L$ in ISTA, ADMM, and ReLU networks. Moreover, we denote by $W^l$ and $W^{l,*}$, $l \in [1, L]$ the weight matrices corresponding to $h \in \mathcal{H}_r$ and $h^*$, respectively. Based on these restricted class of functions, we present the following assumption and theorem (the proof is provided in the supplementary material).

**Assumption 1.** *There exists a constant $C \geq 1$ such that for every probability distribution $\mathcal{D}$, and estimator $h \in \mathcal{H}$, $\mathbb{E}_\mathcal{D} \sum_{j=1}^{n_x} (h_j - h_j^*)^2 \leq C \mathbb{E}_\mathcal{D} \sum_{j=1}^{n_x} \left( \ell(h_j) - \ell(h_j^*) \right)$, where $h_j$ and $h_j^*$ denote the $j$th coordinate of the estimators.*

As pointed out in (Bartlett & Mendelson, 2002), this condition usually follows from a uniform convexity condition on the loss function $\ell$. For instance, if $|h(y) - x| \leq 1$ for any $h \in \mathcal{H}$, $y \in \mathbb{R}^{n_y}$ and $x \in \mathbb{R}^{n_x}$, then the condition is satisfied with $C = 1$ (Yousefi et al., 2018).

**Theorem 5** (Estimation error bound of ISTA, ADMM, and ReLU networks). *Consider the class of functions represented by depth-$L$ ISTA networks $\mathcal{H}_I^L$ as detailed in Section 2.1, $m$ i.i.d training samples, and a per-coordinate loss satisfying Assumption 1 with a constant $C$. Let $\|W^l - W^{l,*}\|_\infty \leq \alpha\sqrt{r}$ for some $\alpha > 0$. Moreover, $B \geq \max\{\alpha\sqrt{r}, 1\}$. Then there exists $T$ in the interval $T \in \left[0, \min\left\{\frac{\sqrt{m} B_0 B^{L-1} 2^L}{\lambda\eta}, m\right\}\right]$, where $\eta = \frac{LB^{L-1}(B-1) - B^L + 1}{(B-1)^2}$, such that for any $s > 0$ with probability at least $1 - e^{-s}$,*

$$\mathcal{E}\left(\mathcal{H}_I^L\right) \leq 41r^* + \frac{17C^2 + 48C}{mn_x} s \tag{13}$$

*where $r^* = C^2\alpha^2 \left( \frac{B_0 B^{L-1} 2^L}{\sqrt{m}} - \frac{\lambda T}{m}\eta \right)^2$. The bound is also satisfied for the class of functions represented by depth-$L$ ADMM networks $\mathcal{H}_A^L$, with $r^* = C^2\alpha^2 \left( \frac{B_0 \tilde{B}^{L-2} 2^{L-1}}{\sqrt{m}} - \frac{\tilde{\lambda} T}{m}\tilde{\eta} \right)^2$, where $\tilde{\lambda} = (1+\gamma)\lambda$, $\tilde{B} = (1+2\gamma)(B+2)$, and $\tilde{\eta} = \frac{(L-1)\tilde{B}^{L-2}(\tilde{B}-1) - \tilde{B}^{L-1} + 1}{(\tilde{B}-1)^2}$, and for the class of functions represented by depth-$L$ ReLU networks $\mathcal{H}_R^L$, with $r^* = C^2\alpha^2 \left( \frac{B_0 B^{L-1} 2^L}{\sqrt{m}} \right)^2$.*

From Theorem 5, we observe that the EE decreases by a factor of $\mathcal{O}\left(1/m\right)$, instead of a factor of $\mathcal{O}\left(1/\sqrt{m}\right)$ obtained for the GE. This result complies with previous local bounds which yield faster convergence rates compared to global bounds (Blanchard et al., 2007; Bartlett et al., 2005). Also, the value of $\alpha$ relates the maximal distance between estimators in $\mathcal{H}_r$ denoted by $r$, to the distance between their corresponding weight matrices $||\boldsymbol{W}^l - \boldsymbol{W}^{l,*}||_\infty \leq \alpha\sqrt{r}$, $l \in [1, L]$. Tighter bounds on the distance between the weight matrices allow us to choose a smaller value for $\alpha$, resulting in smaller values $r^*$ which improve the EE bounds. The value of $\alpha$ could depend on the network's nonlinearity, underlying data distribution $\mathcal{D}$, and number of layers.

Note that the bounds of the model-based architectures depend on the soft-thresholding through the value of $\lambda\mathbb{E}_S(T)$. As $\lambda\mathbb{E}_S(T)$ increases, the bound on the EE decreases, which emphasizes the nonlinearity's role in the network's generalization abilities. Due to the soft-thresholding, ISTA and ADMM networks result in lower EE bounds compared to the bound for ReLU networks. It is interesting to note, that as the number of training samples $m$ increases, the difference between the bounds on the model-based and ReLU networks is less significant. In the EE bounds of model-based networks, the parameter $B_0$ relates the bound to the sparsity level $\rho$, of the target vectors. Lower values of $\rho$ result in lower EE bounds, as demonstrated in the supplementary.

## 5 NUMERICAL EXPERIMENTS

In this section, we present a series of experiments that concentrate on how a particular model-based network (ISTA network) compares to a ReLU network, and showcase the merits of model-based networks. We focus on networks with 10 layers (similar to previous works (Gregor & LeCun, 2010)), to represent realistic model-based network architectures. The networks are trained on a simulated dataset to solve the problem in (1), with target vectors uniformly distributed in $[-1, 1]$. The linear mapping $\boldsymbol{A}$ is constructed from the real part of discrete Fourier transform (DFT) matrix rows (Ong et al., 2019), where the rows are randomly chosen. The sparsity rate is $\rho = 0.15$, and the noise's standard deviation is $0.1$. To train the networks we used the SGD optimizer with the $L_1$ loss over all neurons of the last layer. The target and noise vectors are generated as element wise independently from a uniform distribution ranging in $[-1, 1]$. All results are reproducible through (Authors, 2022) which provides the complete code to execute the experiments presented in this section.

We concentrate on comparing the networks' EE, since in practice, the networks are trained with a finite number of examples. In order to empirically approximate the EE of a class of networks $\mathcal{H}$, we use an empirical approximation of $h^*$ (which satisfy $\mathcal{L}_\mathcal{D}(h^*) = \inf_{h\in\mathcal{H}} \mathcal{L}_\mathcal{D}(h)$) and the ERM $\hat{h}$, denoted by $h^*_{emp}$ and $\hat{h}_{emp}$ respectively. The estimator $h^*_{emp}$ results from the trained network with $10^4$ samples, and the ERM is approximated by a network trained using SGD with $m$ training samples (where $m \leq 10^4$). The empirical EE is given by their difference $\mathcal{L}_\mathcal{D}(\hat{h}_{emp}) - \mathcal{L}_\mathcal{D}(h^*_{emp})$.

In Fig. 2, we compare between the ISTA and ReLU networks, in terms of EE and $L_1$ loss, for networks trained with different number of samples (between 10 and $10^4$ samples). We observe that for small number of training samples, the ISTA network substantially reduces the EE compared to the ReLU network. This can be understood from Theorem 5, which results in lower EE bounds on the ISTA networks compared to ReLU networks, due to the term $\lambda\mathbb{E}_S(T)$. However, for large number of samples the EE of both networks decreases to zero, which is also expected from Theorem 5. This highlights that the contribution of the soft-thresholding nonlinearity to the generalization abilities of the network is more significant for small number of training samples. Throughout the paper, we considered networks with constant bias terms. In this section, we also consider learned bias terms, as detailed in the supplementary material. In Fig. 2, we present the experimental results for networks with constant and learned biases. The experiments indicate that the choice of constant or learned bias is less significant to the EE or the accuracy, compared to the choice of nonlinearity, emphasizing the relevance of the theoretical guarantees. The cases of learned and constant biases have different optimal estimators, as the networks with learned biases have more learned parameters. As a result, it is plausible a network with more learnable parameters (the learnable bias) exhibits a lower estimation error since the corresponding optimal estimator also exhibits a lower estimation error. To analyze the effect of the soft-thresholding value on the generalization abilities of the ISTA network, we show in Fig. 3, the empirical EE for multiple values of $\lambda$. The experimental results demonstrate that for small number of samples, increasing $\lambda$ reduces the EE. As expected from the EE bounds, for a large number of training samples, this dependency on the nonlinearity vanishes, and the EE is similar for all values

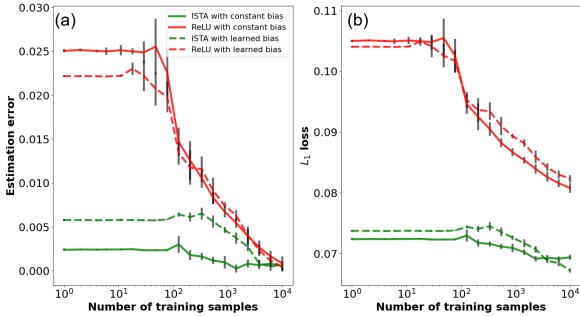

Figure 2: Comparing the EE of ISTA and ReLU networks with 10 layers. **a.** The EE of the networks as a function of the training samples. **b.** The $L_1$ of the networks as a function of the training samples. The ISTA network achieves lower EE compared to the ReLU network, along with lower losses.

of $\lambda$. In Fig. 3b, we show the $L_1$ loss of the ISTA networks for different values of $\lambda$. We observe that low estimation error does not necessarily lead to low loss value. For $m \lesssim 100$, increasing $\lambda$ reduced the EE. These results suggest that given ISTA networks with different values of $\lambda$ such that all networks achieve similar accuracy, the networks with higher values of $\lambda$ provide lower EE. These results are also valid for additional networks' depths. In Section B, we compare the EE for networks with different number of layers, and show that they exhibit a similar behaviour.

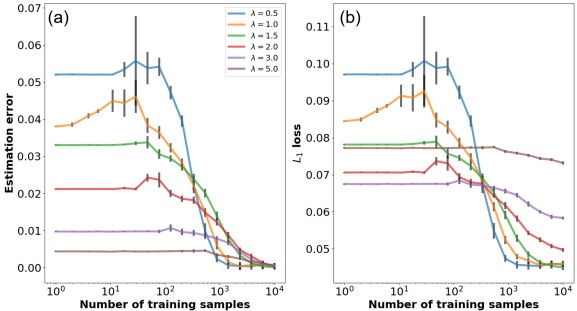

Figure 3: Estimation error and loss of ISTA networks with 10 layers, as a function of the soft-threshold's value $\lambda$. The experiments indicate that the estimation error decreases as $\lambda$ increases, which demonstrates a way to control the networks' generalization abilities through $\lambda$.

## 6 CONCLUSION

We derived new GE and EE bounds for ISTA and ADMM networks, based on the RC and LRC frameworks. Under suitable conditions, the model-based networks' GE is nonincreasing with depth, resulting in a substantial improvement compared to the GE bound of ReLU networks. The EE bounds explain EE behaviours experienced in practice, such as ISTA networks demonstrating higher estimation abilities, compared to ReLU networks, especially for small number of training samples. Through a series of experiments, we show that the generalization abilities of ISTA networks are controlled by the soft-threshold value, and achieve lower EE along with a more accurate recovery compared to ReLU networks which increase the GE and EE with the networks' depths.

It is interesting to consider how the theoretical insights can be harnessed to enforce neural networks with high generalization abilities. One approach is to introduce an additional regularizer during the training process that is rooted in the LRC, penalizing networks with high EE (Yang et al., 2019).

