# OpenReview forum: "Generalization and Estimation Error Bounds for Model-based Neural Networks"
_ICLR.cc/2023/Conference — ICLR 2023 poster_

### Official Review · Reviewer_zELY · 2022-10-19

**Confidence:** 3
**Correctness:** 4
**Technical Novelty And Significance:** 4
**Empirical Novelty And Significance:** 3
**Recommendation:** 8

**Clarity, Quality, Novelty And Reproducibility:**

The paper is clearly written, and besides some minor weak points it seems to respect the common standards of scientific quality, included reproducibility of proofs and numerical experiments.
Up to my limited knowledge of the relevant literature, the paper seems original.

**Details Of Ethics Concerns:**



**Strength And Weaknesses:**

Strong points:
- the bounds that the authors provide seem to be non-vacuous, which I find non-trivial. Indeed, the bounding terms suggests, for example, that the estimation error depends on the soft-thresholding scale parameter, and this is confirmed numerically.
- I could not check the correctness of the proofs given the short time-frame of this review process. Nonetheless, the relevant appendices are very detailed and I feel that the interested reader could check the proof details easily if needed. Also, the results as well as the overall proof techinque seem reasonable.
- The paper is extremely well written and clear. I appreciated particularly that formal mathematical concepts were always explained and justified in the discussion text surrounding them, allowing a much broader audience to understand the results.

Weak points:
- it is not completely clear what role the specific model that the authors consider (eq. 1) plays in the various theorems. This doesn't allow the reader to fully gauge the applicability of the results to different data models.
- (minor weak point) the numerical experiments have been performed fixing the sparsity level of the signal to be retrieved, the noise level and, I suppose, the batch size of SGD. It is not clear whether the numerical results are robust across the choice of these parameters.
- (minor weak point) the numerical experiments mention a "learned bias" setting that is not accurately described in the main text or in the supplementary material.

**Summary Of The Paper:**

The paper studies model-based networks in the context of sparse coding.
A model-based network is constructed by unfolding an iterative algorithm, i.e. mapping each iteration of the original algorithm to a layer of the resulting neural network.
This allows to have a trainable neural network that can adapt to specific problems, while encoding some stuctural information of the original iterative algorithm.

The main goal of the paper is to provide rigorous bounds on the generalisation capabilities of two model-based networks inspired by two specific sparse coding algorithms, namely "iterative shrinkage and thresholding algorithm" (ISTA) and  "alternating direction method of multipliers" (ADMM).
Moreover, the authors aim to compare model-based neural networks to generic fully-connected ReLU networks in the context of these bounds.

The authors prove two sets of bounds for fully-connected ReLU networks and ISTA/ADMM-based neural networks:
- they bound the generalisation error, intended as the gap between empirical and population loss of the worst-performing function in the model class, averaged over training datasets. They find that the bound for model-based networks is better than that of ReLU networks, and this is crucially linked to the effect that component-wise soft-thresholding has on the Rademacher complexity of a given model class.
- they bound the estimation error, intended as the gap between the population loss of the ERM minimiser and the population loss of the best performing function in the model class. Again, model-based networks enjoy a better bound on estimation error than ReLU networks, and again this is linked to the effect of soft-thresholding on the local Rademacher complexity of a given model class.

The authors then study numerically the estimation error of ISTA and ReLU networks, finding that ISTA networks perform better. They also show that the dependence of the estimation error on the scale parameter of the soft-thresholding non-linearity is the same as that predicted by the bounds.

Questions (see also the weak points section):
- how does the specific model considered (eq 1) affect the results?
- are all numerical results robust to a change in sparsity, noise level and batch size? Moreover, what is the batch size used in the numerical experiments?
- the numerical experiments mention a "learned bias" setting that is not accurately described in the main text or in the supplementary material. What is that?

Minor points:
- the authors could add direct links to the proofs of their lemmas and theorems after stating them.
- the statement of assumption 1 is not completely clear to me. Is there some notation change or notation missing?
- if at all relevant, the authors could add the performance of the original ISTA algorithm to their numerical experiments as a baseline performance. If not relevant, why is it not relevant?
- the operative definition of EE (eq 7) is not completely clear. h_hat is the empirical risk minimiser, so it depends on a training set, is this correct? If so, is there some averaging missing?

Curiosities (not relevant for the review process as far as I am concerned):
- is there a natural interpretation to why soft-thresholding lowers the Rademacher complexity of model classes?
- the bound presented in theorem 5 is the same for ReLU, ISTA and ADMM based networks. Can one give a "physical" interpetation to r*? In some sense, r* encodes some information about the network architecture that could determine some other property and not only the EE bound.
- is there an explanation to the fact that learning biases seems to improve the EE of ReLU networks while degrading the EE of ISTA networks?

**Summary Of The Review:**

My initial recommendation is: accept.
The paper is very clearly written, the results are nice, in particular the fact that the bounds obtained seem not vacuous, and, while I did not check the details of the proofs, the theorems seem sound.

---

> ### Author Response · Authors · 2022-11-18
> **Response to reviewer - part 1/2**
>
> >how does the specific model considered (eq 1) affect the results?
> it is not completely clear what role the specific model that the authors consider (eq. 1) plays in the various theorems. This doesn't allow the reader to fully gauge the applicability of the results to different data models.
>
> **Response**
>
> Thank you for your comment, and for helping improving this work.
>
> The considered model affects the EE bound of the network of ISTA network, as different sparsity levels lead to different values of $\lambda$. We analyze the relationship between the bound and the sparsity level in the supplementary material (at the end of Section A.3).
> Although our model-based networks are specific to the model in eq 1, other model-based networks have also been proposed for different inverse problems (e.g. super-resolution problems). We hope that our analysis can also shed some light on why model-based networks also deliver outstanding performance on such inverse problems.
>
> >are all numerical results robust to a change in sparsity, noise level and batch size? Moreover, what is the batch size used in the numerical experiments?
> (minor weak point) the numerical experiments have been performed fixing the sparsity level of the signal to be retrieved, the noise level and, I suppose, the batch size of SGD. It is not clear whether the numerical results are robust across the choice of these parameters.
>
> **Response**
>
> The numerical results are robust for different sparsity levels noise level, dimensions of targets and observations, and number of layers, meaning that small changes in those parameters lead to small changes in the EE and performance of the networks. We provide additional simulation results for neural networks with varying depths in Section B of the supplementary material. During the training process, we used a single batch and performed 150 epochs. The implementation of the networks and the training process are located in an online repository, referenced in the paper.
>
> >the numerical experiments mention a "learned bias" setting that is not accurately described in the main text or in the supplementary material. What is that?
> (minor weak point) the numerical experiments mention a "learned bias" setting that is not accurately described in the main text or in the supplementary material.
>
> **Response**
>
> There are a few versions available for the learned ISTA network. In our work, we consider only one set of learned weight matrices. However, a second set of weight matrices can also be learned, as understood from the following relation between consecutive layers
> $$ \boldsymbol h_I^{l} = S_{\lambda} \left( \boldsymbol W_1^{l } \boldsymbol h_I^{l - 1} + \boldsymbol W_2^{l} \boldsymbol y\right), \ \ \ \boldsymbol h_I^0 = S_{\lambda} (\boldsymbol y).$$
> The comparison that we provide, shows that the main contribution to the lower loss and EE of the ISTA network is due to the soft thresholding nonlinearity, and not the learned bias (the set of additional learned matrices). This emphasizes that the analysis we performed on the ISTA network with constant biases, could be applicable to additional versions of ISTA networks.
> Thank you for this comment. We updated the revised manuscript with a better explanation of the learned biases, in Appendix B of the supplementary material.
>
> **Minor points**
>
> >the statement of assumption 1 is not completely clear to me. Is there some notation change or notation missing?
>
> **Response**
>
> In the assumption, $\boldsymbol h^{\star}$ denotes the optimal estimator, the subscript $j$ represents the $j$th coordinate of the estimator, and $\ell$ is the per coordinate loss defined in eq. 5. We updated the revised manuscript.
>
>
> >if at all relevant, the authors could add the performance of the original ISTA algorithm to their numerical experiments as a baseline performance. If not relevant, why is it not relevant?
>
> **Response**
>
> The focus of this work was to compare the generalization ability of a model-based network to standard ReLU networks, since this is not yet well understood. It has already been established previously that LISTA (a model-based network) outperforms the ISTA algorithm (see e.g. the original paper by Gregor and LeCun [1]) so we did not find it necessary to replicate such results.
>
> >the operative definition of EE (eq 7) is not completely clear. h_hat is the empirical risk minimiser, so it depends on a training set, is this correct? If so, is there some averaging missing?
>
> **Response**
>
> The ERM $\hat{h}$ indeed depends on the training set. The statement is satisfied for any set $S$ containing $m$ samples.

---

> > ### Comment · Reviewer_zELY · 2022-11-21
> > **Questions addressed**
> >
> > I thank the authors for addressing my questions in detail. I have no other concerns regarding the paper.

---

> ### Author Response · Authors · 2022-11-18
> **Response to reviewer - part 2/2**
>
> **Curiosities**
>
> >is there a natural interpretation to why soft-thresholding lowers the Rademacher complexity of model classes?
>
> **Response**
>
> The soft thresholding operation is used in the original ISTA algorithm to shrink the target vector's estimation, due to the $L_1$ regularization in the original problem. This shrinkage operation is also at the heart of the reduction of the RC.
> In practice, the shrinkage should be proportional to the sparsity level of the target vectors, as higher soft thresholding values lead to more sparse targets.
>
> >the bound presented in theorem 5 is the same for ReLU, ISTA and ADMM based networks. Can one give a "physical" interpetation to r*? In some sense, r* encodes some information about the network architecture that could determine some other property and not only the EE bound.
>
> **Response**
>
> Following the LRC framework, $r^*$ is the fixed point of the sub-root function $\psi(r)$, as detailed in Theorem 6. As a result, $r^*$ bounds the RC of the class of functions represented by $\mathcal{H}_r$. Indeed, in the original paper by Bartlett et al. [2], that introduced the LRC, the authors provide additional bounds on the losses of the class of function (in addition to EE bounds). For example, the authors bound the expected loss using empirical loss, based on the value of the fixed point $r^*$.
>
> >is there an explanation to the fact that learning biases seems to improve the EE of ReLU networks while degrading the EE of ISTA networks?
>
> **Response**
>
> When computing the estimation error, we consider estimators in a region close to the optimal estimator. The cases of learned and constant biases have different optimal estimators, as the networks with learned biases have more learned parameters. As a result, it is plausible a network with more learnable parameters (the learnable bias) exhibits a lower estimation error since the corresponding optimal estimator also exhibits a lower estimation error. We discuss this observation in the results section of the revised manuscript, just above Fig. 2.
>
>
> **References**
>
> [1] K. Gregor and Y. LeCun. Learning fast approximations of sparse coding.Int. Conf. Machine Learning, pp. 399–406, 2010.
>
> [2] P.L. Bartlett, O. Bousquet, and S. Mendelson. Local Rademacher complexities. The Annals of Statistics, 33(4):1497–1537, 2005.

---

### Official Review · Reviewer_4ojw · 2022-10-21

**Confidence:** 4
**Correctness:** 4
**Technical Novelty And Significance:** 4
**Empirical Novelty And Significance:** 4
**Recommendation:** 8

**Clarity, Quality, Novelty And Reproducibility:**

(see Strengths and Weaknesses)



**Strength And Weaknesses:**

### Strengths

- The paper provides theoretical insights into the simple (but important) intuition that model-based methods should perform better than model-free approaches. The main ideas and techniques of the work were developed based on well-founded rationales and in general, the paper is well-written with a sufficient literature review.
- The mathematical analyses of the work are rigorous and seem correct. The results of the work, which relies on directly bounding the Rademacher complexity, are significantly stronger and more accessible (readable) than previous works (which focused on estimating Rademacher complexity through covering numbers).
- The bound on Rademacher complexity of soft-thresholding is novel and meaningful and could potentially be of interest to a broader audience in the field beyond the context of model-based networks.
- If the comment below (in the Weakness section) is well addressed, the bound of the Theorem shows that a nonincreasing GE as a function of the network’s depth is achievable, by limiting the weights’ norm in the network. This is an strong improvement over existing bounds, which exhibit a logarithmic increase of the GE with depth.

### Weaknesses

- In my opinion, the paper, in its current state, overstates the significance of its results. I don’t think the results support the claim that their bounds on GE and EE are better than those of ReLU.

    While Lemma 1 establishes that applying soft-thresholding potentially decreases complexity, it doesn’t come with any estimation of the improvement (T) and it is not even clear that T is non-zero. In fact, the statements of the theorems all specify that T’s could be zero. The only place where T is estimated is in the discussion after the proof of Lemma 1 in the Appendix, which states that in certain scenarios, it is possible to bound T from below by an explicit constant depending on m. However, the result is written in a general context and it is unclear how this result applies to the studied networks.

    I strongly suggest that these lower bounds on T are rigorously stated and proved to support the main results of the work. Discussions about the dependence of the improvements on lambda, are not correct if we are not certain that T is non-zero. An explicit bound, even if further assumptions are made, would provide more insights about the order of the improvements, which would be of broad interest to the readers.

- Similarly, in its current state, Lemma 1 is trivial and doesn’t need its complicated proof. To prove its current statement (existence of T in [0, …] such that…), we just need to establish the inequality for T = 0, which can be directly deduced from the contraction lemma (soft-thresholding has Lipschitz constant 1).

### Questions

- In my understanding, the main idea of the proof of Lemma 1 is that although the Lipschitz constant of soft-thresholding is one (it is a translation in certain regions), there are sub-regions of RxR where |S(x) - S(y)| < |x - y| - 2\lambda, and the “improvement” depends on the measure of such sub-regions, specifically, on

    \{ h*(y) > \lambda) \text{and} h*’(y) < -lambda\}

    My question is, the ReLU function also ‘contract’ on this region, and the contraction is at least lambda, i.e., |S(x) - S(y)| < |x - y| - \lambda if x>\lambda and y<-\lambda. Does that also mean that we can obtain a similar result for ReLU using the same argument?

- (Optional. This is not a part of my decision assessment) In the classical proof of the contraction lemma (e.g., Lemma 26.9 of Understanding Machine Learning), the absolute value in the bound can be removed and added when convenient because of the symmetry of h and h’. In the proof of Lemma 1, cases (a, b) is treated differently from (b, a). Is there any particular reason for that?

### Additional feedbacks

- Typos: Section 1.1: “logarithnmic”

**Summary Of The Paper:**

The paper studies generalization and estimation error bounds for model-based neural networks that employ soft-thresholding activation functions. By re-examining the proof of the contraction lemma (of Rademacher complexity) for soft-thresholding functions, the paper shows that applying soft-thresholding potentially decreases complexity (although the Lipschitz constant of soft-thresholding is 1).

Using this result, the paper establishes norm-based bounds on generalization error (GE) and estimation error (EE) of these networks for the problem of sparse vector recovery. The manuscript claims that the GE and EE bounds of the soft-thresholding network are better than those of ReLU. These findings are then validated through experiments.

**Summary Of The Review:**

The paper addresses a meaningful question and the approach is novel and of broad interest. However, the paper, in its current state, overstates the significance of its results. I believe some significant restructuring and additional estimations need to be done to support the claims.

After revision: My main concern about the manuscript have been addressed and I think this is a good paper.

---

> ### Author Response · Authors · 2022-11-18
> **Response to reviewer**
>
> >In my opinion, the paper, in its current state, overstates the significance of its results. I don’t think the results support the claim that their bounds on GE and EE are better than those of ReLU.
> While Lemma 1 establishes that applying soft-thresholding potentially decreases complexity, it doesn’t come with any estimation of the improvement (T) and it is not even clear that T is non-zero. In fact, the statements of the theorems all specify that T’s could be zero. The only place where T is estimated is in the discussion after the proof of Lemma 1 in the Appendix, which states that in certain scenarios, it is possible to bound T from below by an explicit constant depending on m. However, the result is written in a general context and it is unclear how this result applies to the studied networks.
> I strongly suggest that these lower bounds on T are rigorously stated and proved to support the main results of the work. Discussions about the dependence of the improvements on lambda, are not correct if we are not certain that T is non-zero. An explicit bound, even if further assumptions are made, would provide more insights about the order of the improvements, which would be of broad interest to the readers.
>
> **Response**
>
> Thank you for your comments.
> We agree that providing lower bounds on $T$, will give more insights on the presented theorems. In the revision we submitted, we analyzed the value of $T$ and provide a lower bound that is always greater that 0, with some mild assumptions. The lower bound behaves as expected with respect to the networks parameters $B$ and $\lambda$. Increasing $B$, increases the probability that the optimal estimators will cross the threshold in absolute value, and therefore the lower bound on $T$ increases accordingly. Similarly, increasing $\lambda$ makes the event of crossing the threshold less probable, which decreases the lower bound on $T$. Please refer to the discussion just after Theorem 3, for the exact bound. The proof is located in the supplementary material, page 21.
>
> >Similarly, in its current state, Lemma 1 is trivial and doesn’t need its complicated proof. To prove its current statement (existence of T in [0, …] such that…), we just need to establish the inequality for T = 0, which can be directly deduced from the contraction lemma (soft-thresholding has Lipschitz constant 1).
>
> **Response**
>
> We agree with this comment. We have updated Lemma 1 to take into account the value of $T$.
>
>
> >In my understanding, the main idea of the proof of Lemma 1 is that although the Lipschitz constant of soft-thresholding is one (it is a translation in certain regions), there are sub-regions of RxR where |S(x) - S(y)| < |x - y| - 2\lambda, and the “improvement” depends on the measure of such sub-regions, specifically, on
> { h*(y) > \lambda) \text{and} h*’(y) < -lambda}
> My question is, the ReLU function also ‘contract’ on this region, and the contraction is at least lambda, i.e., |S(x) - S(y)| < |x - y| - \lambda if x>\lambda and y<-\lambda. Does that also mean that we can obtain a similar result for ReLU using the same argument?
>
> **Response**
>
> We appreciate this comment. We think that how to re-adapt our proof techniques to RELU based networks will require additional considerations. The reason is our current derivation relies on the symmetry of the non-linearity - i.e. the fact that the nonlinearity is an odd function, this applies to our soft-thresholding nonlinearity induced by unfolding techniques, but it does not apply to RELU. We plan to consider such extensions of our proof technique in our future work.
>
> In any case, the numerical experiments (in our work and in the literature) corroborate that the EE of ISTA networks is lower compared to ReLU networks. As a result, we expect that if a reduction in the RC of ReLU networks is obtained, it will be less significant compared to ISTA networks.
>
> >(Optional. This is not a part of my decision assessment) In the classical proof of the contraction lemma (e.g., Lemma 26.9 of Understanding Machine Learning), the absolute value in the bound can be removed and added when convenient because of the symmetry of h and h’. In the proof of Lemma 1, cases (a, b) is treated differently from (b, a). Is there any particular reason for that?
>
> **Response**
>
> Removing the absolute value when convenient is due to the supremum operation. Due to the symmetry, the supremum interchanges between the estimators if needed. In our proof, we exploit the same concept and show which class of estimator the supremum is achieved. For example, the case (+1,-1) obtains higher RC compared to (-1,+1), as can be understood from the soft thresholding. Instead of relying on the symmetry that neglects the case of (-1, +1), we remove it explicitly.

---

> > ### Comment · Reviewer_4ojw · 2022-11-19
> > **concern addressed**
> >
> > I want to thank the authors for a detailed response and a careful revision of the manuscript. My main concern about the manuscript have been addressed and I think this is a good paper.

---

### Official Review · Reviewer_CF5w · 2022-10-24

**Confidence:** 3
**Correctness:** 4
**Technical Novelty And Significance:** 2
**Empirical Novelty And Significance:** 2
**Recommendation:** 6

**Clarity, Quality, Novelty And Reproducibility:**

The paper is well-organized and clearly written. The results are technically sound and novel.

**Strength And Weaknesses:**

$\textbf{Strength}$

1. It is a novel contribution that broadens the understanding of model-based neural networks and their superiority compared to the standard ReLU networks.

2. The Rademacher complexity of function classes after applying the soft-thresholding operation is derived, which further leads to the generalization and estimation errors of model-based neural networks, such as ISTA and ADMM networks.

3. Codes are provided for the numerical experiments.

$\textbf{Concerns}$

1. The difference between ISTA networks and ReLU networks is that (2) uses a soft-thresholding operator $S_\lambda$ and (10) uses ReLU. Can we think of $S_\lambda$ as a special type of activation function? I was wondering if the performance of ISTA networks is also better than that of leaky ReLU networks or ELU networks?

2. In Figure 3, estimation error decreases as $\lambda$ increases while $L_1$ loss increases as $\lambda$ increases, especially for large sample sizes. In practice, how to choose the best $\lambda$?

3. In Figure 2, it is a little bit confusing to me that the estimation error of ISTA with learned bias is larger than that with constant bias, any explanation?

4.  Is it possible to extend the results to convolutional neural networks?

Minor typos:

1. Is the last term in (13) $\frac{\lambda T^{(l)}}{m}$



**Summary Of The Paper:**

The paper considers the theoretical properties of model-based neural networks which are usually interpretable and inherit the prior structure of the problems, such as ISTA and ADMM networks associated with soft thresholding operators. The bounds for both generalization and estimation errors of these networks are derived via the lens of Rademacher complexity and local Rademacher complexity, which are shown to be lower than that of the common ReLU neural networks under mild conditions. Numerical experiments are also provided to verify the theoretical results.

**Summary Of The Review:**

The paper derives the generalization and estimation error bounds of model-based neural networks and provides insights into the superiority of ISTA and ADMM compared to ReLU networks, which should be of interest to the ICLR community.

---

> ### Author Response · Authors · 2022-11-18
> **Response to reviewer**
>
> >The difference between ISTA networks and ReLU networks is that (2) uses a soft-thresholding operator Sλ  and (10) uses ReLU. Can we think of  Sλ  as a special type of activation function? I was wondering if the performance of ISTA networks is also better than that of leaky ReLU networks or RELU networks?
>
> **Response**
>
> Thank you for your comments, and help in improving this work!
>
> In fact, the soft-thresholding operator can be obtained by subtracting two shifted ReLU functions. We show that this difference in the activation of the network (that arises from the ISTA model) significantly affects the network's generalization ability. A similar analysis can be performed on additional activation functions to better understand their role in the generalization ability of neural networks. Although it is not immediate, GE and EE bounds for neural networks with leaky ReLU activations could be obtained with the contraction lemma, as the Lipschitz constant of the activation (which corresponds to the slope of the leaky ReLU). However, we did not carry out this analysis because our focus was on comparing well-known model-based networks to the most common ReLU-based neural network.
>
> >In Figure 3, estimation error decreases as λ increases while L1 loss increases as λ increases, especially for large sample sizes. In practice, how to choose the best λ?
>
> **Response**
>
> Following the original model of ISTA, $\lambda$ is a hyperparameter of the optimization problem. The optimal value of $\lambda$ generally depends on the target vector sparsity level. Note however that for learned networks, the value of $\lambda$ at each layer can also be learned, which increases the degree of freedom learned by the network. Our analysis applies for simplicity for a fixed value of $\lambda$. We added a discussion on the choice of $\lambda$ in Section 2.1, which describes the network architectures.
>
> >In Figure 2, it is a little bit confusing to me that the estimation error of ISTA with learned bias is larger than that with constant bias, any explanation?
>
> **Response**
>
> When computing the estimation error, we consider estimators in a region close to the optimal estimator. The cases of learned and constant biases have different optimal estimators, as the networks with learned biases have more learned parameters. As a result, it is plausible a network with more learnable parameters (the learnable bias) exhibits a lower estimation error since the corresponding optimal estimator also exhibits a lower estimation error. We discuss this observation in the results section of the revised manuscript, just above Fig. 2.
>
> >Is it possible to extend the results to convolutional neural networks?
>
> **Response**
>
> Thank you for this very interesting comment. Yes, it is possible. The analysis of the soft thresholding operation still holds. Also, as the convolution operation can be expressed via multiplication with a convolution matrix, the results we present can be extended relatively easily to convolutional neural networks. We added this comment to the revised version after the discussion of Theorem 3, as this might be of interest to potential readers.

---

### Official Review · Reviewer_FqeZ · 2022-10-25

**Confidence:** 2
**Correctness:** 3
**Technical Novelty And Significance:** 3
**Empirical Novelty And Significance:** 3
**Recommendation:** 6

**Clarity, Quality, Novelty And Reproducibility:**

* The paper is very clear. Very organized and enjoyable to read.

* The paper is of high quality.

* The paper is quite novel to me.

**Strength And Weaknesses:**

**Strength**

* I am not an expert on model-based neural network, but from my perspective, I think this work adopts a suitable theoretical tools to solve an important problem.

* This writing is quite good. It is eazy to follow the main idea of this work.

**Weakness**

* Though the authors provide simulations to verify the proposed theory, it is still not clear how realistic the theory is and what is the gap between the considered scenario and the realistic settings.

**Summary Of The Paper:**

This work studies the generalisation properties of model-based neural networks which can achieve unparalleled performance on sparse coding and compressed sensing problems. In particular, the authors leverage complexity measures including the global and local Rademacher complexities to provide upper bounds on the generalisation and estimation errors of model-based networks. They show that the generalisation abilities of model-based networks for sparse recovery outperform those of regular ReLU networks.

**Summary Of The Review:**

I understand the context of this work and the results it is trying to convey. If the results of this work are correct, I think this work deserves to be accepted by ICLR.

---

> ### Author Response · Authors · 2022-11-18
> **Response to reviewer**
>
> >Though the authors provide simulations to verify the proposed theory, it is still not clear how realistic the theory is and what is the gap between the considered scenario and the realistic settings.
>
> **Response**
>
> Thank you for your comments.
> This work shows that ISTA and ADMM networks provide a reduction in the GE and EE compared to ReLU networks. This reduction depends on the network architecture (depth, bound on the norm of the weight matrices), and the underlying data distribution $\mathcal{D}$. In the revision we submitted, we analyzed the value of $T$ and provide a lower bound that is always greater than 0, with some mild assumptions. The lower bound behaves as expected with respect to the networks parameters $B$ and $\lambda$. Increasing $B$, increases the probability that the optimal estimators will cross the threshold in absolute value, and therefore the lower bound on $T$ increases accordingly. Similarly, increasing $\lambda$ makes the event of crossing the threshold less probable, which decreases the lower bound on $T$. Please refer to the discussion just after Theorem 3, for the exact bound. The proof is located in the supplementary material, page 21.
>
> This analysis allows us to establish that a model-based network generalizes better than traditional ReLU feedforward neural networks. Our simulations corroborate our analysis since they show that model-based networks indeed generalize better than traditional ones.

---

### Decision · Program_Chairs · 2023-01-20

**Decision:**

Accept: poster

**Justification For Why Not Higher Score:**

While this is a very nice paper I did not see any particular reason for a highlight. But would not be against either.



**Justification For Why Not Lower Score:**

No reason to reject.



**Metareview: Summary, Strengths And Weaknesses:**

All reviewers agree this paper should be accepted. I think the reviews summarize very well the strengths and weaknesses of the paper as well as points that the authors should include in the revised version. I think this will be a great addition to the conference

**Note From Pc:**

if the above contains the word "oral" or "spotlight" please see: "oral" presentation means -> notable-top-5% and "spotlight" means -> notable-top-25%. As stated in our emails, we are disassociating presentation type from AC recommendations